# Integrative Epigenetic and Molecular Analysis Reveals a Novel Promoter for a New Isoform of the Transcription Factor TEAD4

**DOI:** 10.3390/ijms25042223

**Published:** 2024-02-13

**Authors:** Shima Rashidiani, Gizaw Mamo, Benjámin Farkas, András Szabadi, Bálint Farkas, Veronika Uszkai, András Császár, Barbara Brandt, Kálmán Kovács, Marianna Pap, Tibor A. Rauch

**Affiliations:** 1Institute of Biochemistry and Medical Chemistry, Medical School, University of Pécs, 7624 Pécs, Hungary; sh.rashidiani@gmail.com (S.R.); mamo_gizaw@yahoo.com (G.M.); benjamin.farkas@aok.pte.hu (B.F.); szabadi.andras@pte.hu (A.S.); 2Department of Dentistry, Oral and Maxillofacial Surgery, Medical School, University of Pécs, 7623 Pécs, Hungary; 3Department of Obstetrics and Gynecology, Medical School, University of Pécs, 7624 Pécs, Hungary; farkas.balint@gmail.com (B.F.); uszkai.veronika@pte.hu (V.U.); csaszar.andras@pte.hu (A.C.);; 4National Laboratory of Human Reproduction, University of Pécs, 7624 Pécs, Hungary; 5Department of Medical Biology and Central Electron Microscope Laboratory, Medical School, University of Pécs, 7624 Pécs, Hungarymarianna.pap@aok.pte.hu (M.P.)

**Keywords:** TEAD4, Hippo/TEAD signaling, alternative promoter, transcriptional regulation, DNA methylation

## Abstract

TEAD4 is a transcription factor that plays a crucial role in the Hippo pathway by regulating the expression of genes related to proliferation and apoptosis. It is also involved in the maintenance and differentiation of the trophectoderm during pre- and post-implantation embryonic development. An alternative promoter for the TEAD4 gene was identified through epigenetic profile analysis, and a new transcript from the intronic region of TEAD4 was discovered using the 5’RACE method. The transcript of the novel promoter encodes a TEAD4 isoform (TEAD4-ΔN) that lacks the DNA-binding domain but retains the C-terminal protein–protein interaction domain. Gene expression studies, including end-point PCR and Western blotting, showed that full-length TEAD4 was present in all investigated tissues. However, TEAD4-ΔN was only detectable in certain cell types. The TEAD4-ΔN promoter is conserved throughout evolution and demonstrates transcriptional activity in transient-expression experiments. Our study reveals that TEAD4 interacts with the alternative promoter and increases the expression of the truncated isoform. DNA methylation plays a crucial function in the restricted expression of the TEAD4-ΔN isoform in specific tissues, including the umbilical cord and the placenta. The data presented indicate that the DNA-methylation status of the TEAD4-ΔN promoter plays a critical role in regulating organ size, cancer development, and placenta differentiation.

## 1. Introduction

Although organ-size coordination, tumor growth regulation, and trophectoderm differentiation are seemingly completely different cellular processes, they are linked by at least one common regulatory network, the TEAD/Hippo pathway. Originally identified in Drosophila in a screen for tissue growth regulators [1], it has since been shown that all essential components of the TEAD/Hippo regulatory cascade are present in mammals [2]. The TEAD/Hippo pathway comprises an intricate network of more than 30 core elements, including ligands, receptors, protein kinases, transcription factors, and transcriptional cofactors [3]. TEA-domain transcription factors (TEADs) belong to the transcription-enhancer factor (TEF) family, which has the TEA/ATTS DNA-binding domain and recognizes the TGGAATGT consensus sequence in promoter regions [4]. The four TEAD proteins exhibit a high degree of homology, with a range of 61% to 73%, and possess a DNA-binding domain at the N-terminus and a YAP/TAZ-binding domain at the C-terminus. Nearly all tissues express at least one of the TEAD genes, and some express all four [5,6]. TEAD1 enhances the expression of genes specific to the heart and is believed to be critical for myocardial differentiation [7]. TEAD2 is involved in regulating gene expression during neural development [8]. The precise function of TEAD3 has yet to be elucidated. The regulatory function of TEAD4 was initially explored in the context of embryo implantation, unveiling its involvement in the differentiation of blastomeres into the trophectoderm [9,10]. In Tead4 mutants, there is a substantial decrease in Cdx2 expression in blastomeres, which then differentiate into inner-cell-mass cells, implying that Tead4 plays a critical role in the initiation of Cdx2 expression, a crucial gene in TE development [9,11]. The absence of TEAD4 results in lowered mitochondrial activity and heightened levels of oxygen-reactive species in pre-implantation mouse embryos [12]. Additionally, TEAD4 is detected in trophoblast stem-cell-like progenitor cells (TSPCs), and the loss of Tead4 in post-implantation mouse TSPCs impairs their self-renewal, resulting in embryonic lethality before 9.0 days of embryonic development, which corresponds to the first trimester of human pregnancy [13]. An accurate comprehension of the mechanism of action of the TEAD/Hippo cascade has been disclosed in the tumor context. Consequently, Hippo-signaling core kinases remain inactive at low cell densities, leading to unphosphorylated YAP1’s translocation into the nucleus, where it interacts with TEAD4 [14]. The binding of TEAD4-YAP1 triggers cell proliferation through the activation of cell-division-promoting genes and anti-apoptotic genes. In contrast, when cells reach a point of contact-mediated inhibition (CMI), upstream modulators of the Hippo pathway, like E-cadherin, are activated, leading to the phosphorylation of YAP1. Phosphorylated YAP1 becomes degraded in the cytoplasm, eventually resulting in cell proliferation inhibition [15,16,17]. The TEAD/Hippo pathway undergoes downregulation during tumorigenesis, resulting in uncontrolled cell growth and tumor-cell metastasis. Recent genome-wide studies demonstrate that the association between DNA methylation and gene expression is more intricate than previously understood and is dependent on the specific genomic region involved (such as the promoter or intragenic region), which is often linked to the gene’s epigenetic context [18]. DNA methylation is linked to reduced transcriptional activity in the promoter region, but highly methylated intragenic (i.e., intronic) areas are associated with elevated transcriptional rates [19,20]. Methylation, or its lack thereof, may determine the transcriptional activity of alternative promoters [21,22]. DNA methylation is not the only mechanism involved in the regulation of intragenic promoters, but it is associated with a specific epigenetic histone signal (e.g., H3K36me3). DNA and histone hypomethylation promote transcriptional activity of alternative promoters during tissue- and developmental-stage-specific gene expression [21,23,24]. A comprehensive epigenetic reprogramming takes place during early embryonic development, erasing the parental DNA methylation pattern and creating a new one with profound implications for the segregation of the trophectoderm and inner cell mass [10]. This critical change in the DNA methylation pattern has the potential to alter the patterns of gene and isoform expression of TEAD4.

Our study reveals that ChIP-Seq data analysis can predict an alternative promoter for the TEAD4 gene. Consequently, a previously unknown transcript is initiated from an intronic region of TEAD4 and is expressed only in certain tissues. The newly identified TEAD4 transcript encodes a truncated isoform (TEAD4-ΔN) that lacks the DNA-binding domain and is mainly localized in the cytoplasm. The alternative promoter of TEAD4 exhibits differential methylation between expressing and non-expressing cell types, suggesting a strict epigenetic control of isoform expression. Furthermore, TEAD4 interacts with the alternative promoter region, which results in the upregulation of TEAD4-ΔN expression. DNA-methylation-mediated epigenetic regulation of the novel promoter may be highly relevant in biological and pathological contexts (i.e., early mammalian development and tumorigenesis) where DNA methylation plays a critical role in controlling genetic reprogramming and cancer-specific gene expression.

## 2. Results

### 2.1. Identification of a Novel Promoter for TEAD4

Major genetic/genomic databases, such as NCBI, ENCODE, and Ensembl, list only a single TEAD4 gene without any additional promoter(s) that produce alternative isoform(s). Typically, transcription factors with complex regulatory networks, like TEAD4, have numerous isoforms [21,25,26]. Our hypothesis was that the TEAD4 gene may also have alternative promoter(s) and that corresponding transcript(s) could play a role in fulfilling its intricate regulatory function. To identify novel and unexplored TEAD4 gene promoters, we analyzed the epigenetic landscape of the TEAD4 gene and its surrounding chromosomal regions using data from the ENCODE databases (Figure 1). Initially, we focused on the epigenetic histone signals that define the transcriptionally active promoter regions. For instance, there is strong evidence that the tri-methylation of histone H3 at lysine 4 (H3K4me3), the acetylation of histone H3 at lysine 27 (H3K27ac), and the deposition of the Z isoform of histone 2A (H2A.Z) are signals of transcription initiation. In addition, the presence of a transcriptionally active promoter is highly indicated if the RNA polymerase 2 (Pol2) signal peak aligns with these signals. Furthermore, the ENCODE database contains ChIP-Seq datasets for various transcription factors, which demonstrate that the intronic region is heavily enriched in certain regions that overlap with histone signals defining active promoters (Appendix A). Examination of these signals was conducted on two cell lines, K562, a human myelogenous leukemia cell line, and H1-hESC, a totipotent human embryonic cell line, both of which are well-characterized. The analysis focused on the H1-hESC and K562 cell lines due to the ENCODE database revealing that the intronic region of the TEAD4 gene has the most characteristic epigenetic milieu to support transcriptional competence. Two regions displayed overlapping epigenetic signals, implying substantial transcriptional potential. One region was the well-known canonical TEAD4 promoter, and the other was located ~40 kbs downstream of the canonical promoter in intron 3. The predicted promoter, which has not yet been explored, can only be functional in specific cell types, as evidenced by ChIP-Seq data. This implies that the corresponding transcript initiated by this alternative TEAD4 promoter might have a distinct role.

### 2.2. Identification of a Novel TEAD4-Isoform-Encoding Transcript

In order to determine the transcriptional activity and pinpoint the transcription start site (TSS) of the putative promoter, a 5’RACE experiment was conducted using total RNA extracted from K562 cells, which has been validated for transcriptional competence in previous in silico investigations (Figure 1). The obtained fragment was subsequently PCR-amplified (Figure 2a, lane 1), cloned into a suitable cloning vector, and subjected to Sanger sequencing to precisely identify the TSS. After aligning the nucleotide sequence of the cloned fragments with the human genome, we identified a novel TSS within intron 3. It is noteworthy that this TSS coincided with the genomic region anticipated by epigenetic signals (Figure 1). To generate transcript-specific primer pairs, we employed this knowledge and designed a forward PCR primer that binds to the alternative exon and a reverse primer positioned in the 3’UTR. After amplification (Figure 2a, lane 2), the fragment was cloned and sequenced using Sanger sequencing. Analysis of the data suggests that the exon identified by the new promoter is non-coding, whereas the entire transcript generates a truncated form of the TEAD4 protein (designated as TEAD4-DN). Interestingly, the C-terminal region of the truncated protein is identical to the full-length TEAD4 protein.

### 2.3. The Novel TEAD4 Isoform Encodes a DNA-Binding-Domainless Protein

In the next step, the coding capacity of the novel TEAD4 mRNA variant was investigated. We detected an open reading frame that encodes a truncated TEAD4 isoform lacking a DNA-binding domain (Figure 3). Consequently, the absence of the TEA/ATTS DNA-binding domain results in compromised DNA-binding capacity of the encoded TEAD4 isoform. This therefore suggests that this isoform may have a unique cellular function.

### 2.4. The TEAD-ΔN Isoform Is Excluded from the Nucleus

Several mechanisms have been described for regulating the subcellular localization of proteins. For instance, the nuclear localization signal (NLS) of NF-κB is covered by specific inhibitory proteins that prevent its constitutive entry into the nucleus [27]. However, proteins with no NLS can still fulfill their regulatory function through NLS-independent nuclear localization in cells [28]. The removal of the DNA-binding domain poses an interesting question about the cellular and molecular function of the TEAD-ΔN isoform. NLS prediction software (NLStradamus 1.0) could not detect this kind of sequence in the truncated TEAD4 isoform. Experimental evidence was sought to determine the location of the isoform. Our hypothesis was that identifying the subcellular localization of this isoform could provide insight into its potential function. Therefore, both isoforms were cloned into plasmids encoding fluorescent proteins. The full-length TEAD4 isoform (TEAD4-FL) was cloned into a plasmid expressing GFP (green fluorescent protein), while TEAD4-ΔN was cloned into a plasmid expressing RFP (red fluorescent protein). The recombinant plasmid constructs were transiently co-transfected into eukaryotic cells. The full-length isoform of TEAD4 (TEAD4-FL) was found exclusively in the nuclei, whereas the truncated form (TEAD4-ΔN) was mainly present in the cytoplasm (Figure 4). The cytoplasmic location of the isoform with a truncated DNA-binding domain may affect the regulation of the Hippo signaling pathway, as discussed later.

### 2.5. TEAD4-ΔN Expression Is Cell-Type-Specific

Cell-type-specific gene expression of the TEAD4 isoforms was investigated in total RNA samples isolated from various normal human tissues (Figure 5). Isoform-specific end-point PCR was used to investigate the gene expression of both isoforms in parallel. The results showed that the full-length isoform of TEAD4 was expressed in all the investigated tissue samples. In contrast, TEAD4-ΔN gene expression was found to be specific to certain cell types, suggesting a more intricate role for this novel isoform.

Western blotting was performed to determine the protein expression pattern of the TEAD4 isoforms in stable cell lines (Figure 6). Like the PCR studies, we detected the long isoform in all tested cell lines, whereas the short isoform was only present in specific cell lines, with significantly lower expression levels compared with the long isoform. The RNA- and protein-based data demonstrate isoform-specific differences in expression levels and distribution for the truncated TEAD4 isoform.

### 2.6. In Vitro Analysis of the Alternative TEAD4 Promoter

Transcriptional regulation is based on the combinatorial binding of transcription factors (TFs) in the promoter and enhancer regions [29]. These bindings promote the formation of pre-initiation complexes and subsequent efficient transcription by the Pol2 enzyme. A search for TF binding sites in the TEAD4-ΔN promoter uncovered multiple potential binding sequences, including a consensus motif for TEAD4 itself, suggesting an interesting self-regulatory mechanism. To confirm that the in silico predicted TEAD4 consensus (in the TEAD4-ΔN promoter) binds to this cis element, electrophoretic mobility-shift assays (EMSAs) were performed initially (Figure 7). As there were no commercially available high-quality antibodies for TEAD4 chips, we utilized a competitive EMSA approach to confirm TEAD4’s binding to the novel promoter region.

The nucleotide sequence of the predicted cis-element for TEAD4 binding is very similar to the consensus TEAD4 sequence [4] (Figure 7a). However, the core six nucleotides of the binding site are identical, differing in only one flanking nucleotide. A comparison of lanes 2 and 7 in Figure 7b demonstrates the binding efficency between the consensus and the actual TEAD4-ΔN promoter-related motifs. Accordingly, two TEAD4-specific bands were detected (labeled with asterisks) in EMSAs. The EMSA studies demonstrate that band B is exclusively formed by TEAD4 since the mutated promoter element lacks this band (Figure 7, lane 5), and competition with TEAD4 consensus oligonucleotide successfully eliminated band B (Figure 7, lane 4). TEAD4 might also be involved in the formation of band A; however, other interacting TFs are also shown to play a role in this DNA–protein complex formation.

### 2.7. Functional Characterization of TEAD4 Promoter(s) in Transient Transfection Studies

To gain more insight into the transcriptional regulation of TEAD4-ΔN expression and to evaluate the validity of the in vitro DNA–protein interaction studies, the promoter region upstream of the transcription start site (TSS) was amplified by PCR, and the obtained fragment was inserted into an expression vector upstream of the luciferase reporter gene. The 1.3-kilobase intron region was selected based on Ensembl Regulatory Build [30], a comprehensive database of epigenetic markers and transcription factors that provided a concise summary of potential TEAD4-ΔN regulatory regions (Appendix A). Additionally, the evolutionary conservation of this region suggests a complex role in transcriptional regulation (Appendix A). Both the canonical TEAD4 and the TEAD4-ΔN promoters were cloned into recombinant luciferase-reporter-gene-containing plasmids. These plasmids were then transiently transfected into HEK293 cells to evaluate their activity in a eukaryotic milieu, as shown in Figure 8. In the assays conducted, the recently discovered promoter exhibited luciferase activity that was significant but less pronounced than the activity measured for the canonical promoter. This suggests that the isoform containing the DNA-binding domain may have the dominant cellular function, while the other isoform (TEAD4-ΔN) may only be necessary under specific conditions and in certain cellular environments.

In vitro studies have shown that the TEAD4 transcription factor can engage the TEAD4 cis-element in the novel intronic promoter (Figure 7b). We assessed the impact of TEAD4 binding to this promoter region by co-transfecting a TEAD4-overexpressing plasmid with the reporter-gene construct carrying the 1.3 kb long TEAD4-ΔN promoter. By co-transfecting increasing amounts of a TEAD4-overexpressing plasmid, the promoter activity gradually increased, as demonstrated by increased luciferase activity. This implies that TEAD4 is a positive regulator of the expression of the truncated isoform of TEAD4 (Figure 8b). It is notable that high levels of TEAD4 can be inhibitory for promoter activity. Presented data sets in vitro and in vivo demonstrate that TEAD4 regulates the expression of its own isoform by interacting with its novel intronic promoter region.

### 2.8. TEAD4-ΔN Expression in Human Placenta Is Regulated by DNA Methylation

TEAD4 ensures the survival of human embryos after implantation by regulating the self-renewal and development of trophoblast progenitors in the placental primordium [13]. Research suggests that loss of TEAD4 in post-implantation TSPCs of mice diminishes their ability to self-renew, resulting in embryonic lethality before embryonic day 9.0, a developmental stage corresponding to the first trimester of pregnancy in humans [13]. Therefore, it is of great interest to determine how the expression of TEAD4-ΔN is involved in this developmental process. As an initial investigation, we analyzed the expression pattern of the TEAD4 isoforms in samples of human placenta and umbilical cord tissue (Figure 9a). We observed that the expression of the two TEAD4 isoforms was different in the analyzed human samples. Specifically, TEAD4-ΔN was not detected in the umbilical cord samples, while it was present in the placental lysates.

Epigenetic signals including DNA methylation and various histone modifications generate a chromatin milieu that sets the stage for efficient transcription. As a next step, we examined the probable epigenetic regulatory mechanisms underlying selective TEAD4 isoform expression. We conducted DNA methylation studies, including bisulfite sequencing, on genomic DNA samples isolated from umbilical cords and placentas. Our epigenetic studies were focused on revealing the DNA methylation status of the two TEAD4 promoters (i.e., the canonical and the alternative one), which might provide an explanation for the observed protein expression differences. The bisulfite sequencing data is consistent with the observed protein expression pattern. The canonical TEAD4 promoter is not methylated in either the umbilical or placental samples, allowing for uniform TEAD4 protein expression in these tissues. The promoter of the truncated isoform is heavily methylated in umbilical samples, resulting in impaired transcription, and, accordingly, there is no expression of the TEAD4-ΔN isoform in such samples, explaining the protein expression pattern detected by Western blotting.

## 3. Discussion

A new TEAD4-isoform-encoding transcript was identified, which is initiated from an alternative intronic promoter region. The TEAD4-ΔN transcript encodes an N-terminus truncated version of the well-characterized TEAD4 transcription factor. The novel transcript does not encode the DNA-binding domain (i.e., TEA/ATTS), suggesting that it is not directly involved in transcriptional regulation. This conclusion is supported by the observation that the TEAD4-ΔN:RFP chimera protein has no NLS and can be detected predominantly in the cytoplasm of transfected cells. A publication describes a short TEAD4 isoform, which is attributed to an alternative splicing event [31]. According to this report, the generation of the short TEAD4 isoform is associated with tumor-associated exon skipping, which is a rather rare event. We attempted to detect the corresponding mRNA produced by the alternative splicing event in various cell types but were unable to do so. However, we demonstrate that alternative promoter usage can also generate this variant with higher frequency, and it is not necessarily an erroneous splicing event associated with tumor formation. The TEAD4-ΔN transcript encodes the full-length YAP-binding domain, which may interact with the YAP1 protein in the cytoplasm and interfere with it. Phosphorylation of YAP-1 results in its translocation into the nucleus. Further investigation into TEAD4-ΔN-mediated regulation of the Hippo pathway may provide additional information on the finely-tuned mechanism involved in the differentiation of cell lines that affect embryo implantation in the uterus and placenta formation. Our study demonstrates that the expression of the TEAD4-ΔN-isoform-encoding mRNA is dependent on the DNA methylation status of the intronic promoter (Figure 10). The absence of DNA methylation may create a chromatin environment that promotes the decondensation of nucleosomes and attracts the binding of various transcription factors, including the DNA-binding-domain-harboring TEAD4. TEAD4 can form heterodimers with TEAD family proteins and SMAD TFs as well, increasing its significance in gene regulation [32]. During the pre- and post-implantation period of embryonic life, DNA-methylation-based epigenetic regulation is essential [33,34]. The DNMT3B enzyme is primarily responsible for de novo DNA methylation, which is necessary for regulating placental development and function [35]. The regulation of TEAD4-ΔN expression by DNA methylation raises intriguing questions about the molecular function of the truncated TEAD4 isoform. It is currently known that TEAD4 lacking a DNA-binding domain can disrupt the Hippo–YAP signaling pathway and interfere with cell proliferation, cell migration, and organ growth [31]. A short isoform of TEAD4 due to alternative splicing, known as TEAD4-S, has been reported [31]. It is highly possible that this isoform is identical to our TEAD4-ΔN at the protein level. In vitro studies have shown that TEAD4-S can inhibit the translocation of YAP to the nucleus and impair its interaction with transcription factors, including TEAD4. This can lead to remodeling of the whole transcriptome and disruption during tumorigenesis. We demonstrate here that occurrence of the short TEAD4 isoform is not necessarily associated with tumorigenesis; it can be expressed in a cell-type-specific manner from the newly discovered promoter as well. It is well known that DNA methylation can be heavily involved in the regulation of cell-type-specific gene expression [36], and in the preimplantation period of embryonic life, paternal and maternal DNA methylation patterns are erased and newly established [37]. Here, we have described a new TEAD4 isoform, the expression of which is tightly regulated by DNA methylation. According to a report, the short isoform of TEAD4 may impede TEAD4-mediated gene activation in cancer cell lines [31]. We observed that the promoter region of TEAD4 could be heavily methylated. Therefore, the use of DNA methyltransferase inhibitors (such as decitabine or azacitidine) may be a potential treatment option for certain tumor types [38]. In some tumors, TEAD4-ΔN may be overexpressed, making its silencing potentially beneficial. Treatment with DNA demethylase inhibitors, such as TET inhibitors, could be advisable [39]. Therefore, a systematic analysis of the DNA methylation status of the TEAD4-ΔN promoter in tumor samples can have diagnostic significance and provide guidance for treatment options. 

Information on TEAD4-ΔN expression in preimplantation embryos (i.e., blastomeres) is currently unavailable. Therefore, it is unclear how TEAD4-ΔN expression can affect subsequent regulatory mechanisms. Although there are significant differences between human and mouse ontogenesis, employing an animal model can shed light on basic processes [40]. Accordingly, the evolutionary conservation of the TEAD4 promoter allows for the investigation of these processes in mice. TEAD4 and YAP are also involved in stem-cell renewal, which is essential during placenta development and for sustaining a functional placenta. Targeting TEAD4-ΔN in endometrial implantation and the subsequent placenta formation could have practical therapeutic significance, which might differ in an oncology context. This is because current DNA methyltransferase inhibitors and TET inhibitors may cause mutations that lead to malformations during embryonic development. A new generation of DNA methylation inhibitors is on the horizon, which may increase their potential use for implantation-related issues [38] in the future.

The molecular function of TEAD4-ΔN and its interacting partner proteins is not yet fully understood. Further research is needed to reveal the underlying mechanisms, which could have even more therapeutic and diagnostic implications.

## 4. Materials and Methods

### 4.1. Epigenetic Data Analysis

Histone-mark-related chromatin-immunoprecipitation data sets (i.e., ChIP-Seq data) were obtained from the ENCODE database (access date: 12.12.2023) [41]. The University of California Santa Clara, CA, USA maintains a publicly available source of ChIP-Seq research data.

### 4.2. Cell Culturing

The K562 (ATTC CCL-243), HEK293T (ATCC CTL-3216), and glioblastoma cell lines were cultured in DMEM medium supplemented with 10% fetal bovine serum. The cell cultures were maintained in a humidified incubator at 37 °C with 5% CO_2_ in air.

### 4.3. Total RNA Isolation

Total RNA was prepared from cell cultures using the Direct-zol RNA miniprep kit (Zymo Research, Irvine, CA, USA). RNA preparation from human samples was approved by the Ethics Committee of the University of Pécs (Code: 3648—PTE 2020, Epigenetic and Transcriptional Factors Involved in Placental Development). Additional total RNA samples used in this study were obtained from the FirstChoice^®^ Human Total RNA Survey Panel (ThermoFisher Scientific, Waltham, MA, USA). 

### 4.4. 5′RACE (5′ Rapid Amplification of cDNA Ends)

The 5′RACE method was utilized to determine the transcription start site of the uncharacterized RNA transcript [26]. 5′RACE was performed on 1 μg of total RNA isolated from K562 cells, resulting in a cDNA copy of the RNA sequence of interest. The 5′ end was amplified using anchor- and gene-specific primers, and the resulting fragment was visualized on an agarose gel. Subsequently, the amplified fragment was cloned into the pDrive plasmid (Qiagen, Hilden, Germany), and the nucleotide sequence was determined by Sanger sequencing. The second-generation 5′/3′ RACE kit (Sigma-Aldrich, Saint Louis, MO, USA) was used to identify the start site of the TEAD4-ΔN-encoding mRNA transcript.

### 4.5. Nucleotide Sequence Analysis

Recombinant plasmids purified from bacteria were sequenced (Sanger sequencing) at the Department of Medical Genetics (University of Pécs), and “DNA Blat v23” software was used to identify the corresponding human genomic region.

### 4.6. Confocal Fluorescent Microscopy

To investigate the cellular localization of the two TEAD4 isoforms, we synthesized the corresponding ORFs in vitro (IDT, Coralville, IA, USA) and cloned them in-frame with green fluorescent protein (pEGFP-N1)- or red fluorescent protein (pDsRED-monomer-N1)-expressing mammalian expression vectors. We verified the correct fusions of the coding regions using Sanger sequencing. Recombinant plasmids carrying the full-length TEAD4-FL-GFP isoform and the N-terminal-truncated TEAD4-ΔN isoform were purified from bacteria and used in transient co-transfection studies. HEK293T cells were plated 18–24 h prior to transfection, and cell cultures were required to be at least 80% confluent at the time of transfection. The plasmids were combined in a 1:1 ratio, and the 293Tran transfection reagent (OriGene Technologies, Rockville, MD, USA) was used for co-transfection. After 48 h of transfection, HEK293T cells were fixed with paraformaldehyde and their nuclei were stained with 4,6-diamidino-2-phenylindole. The co-transfection efficiency was approximately 30%. Images were captured using a Zeiss LSM 700 confocal microscope and analyzed with Zen 2. software v2.2 (Zeiss). The slides were examined at a magnification of 800×.

### 4.7. Luciferase Reporter Assay

The promoter regions of TEAD4 and TEAD4-ΔN were amplified by PCR and cloned upstream of the luciferase reporter gene into the XhoI-HindIII sites of pGL3-basic plasmids. The recombinant plasmids were purified from bacteria using the ZymoPURE—Express Plasmid Midiprep kit (Zymo Research, Irvine, CA, USA), and 1 μg was transfected into HEK293 cells using the GenJet™ in vitro DNA transfection reagent. Cells that were transfected were harvested 48 h later, and luciferase activity was measured using the ONE-Glo™ Luciferase Assay System (Promega, Madison, WI, USA). The luciferase activity was normalized to the protein content, and the relative fold-change was calculated by considering the measured luciferase activity to be 1 in empty pGL3-basic-transfected samples. The untagged TEAD4-expressing plasmid was purchased from Origene (Rockville, MD, USA). Co-transfection was performed using 1, 10, 50, 100, and 250 ng of plasmid along with 1 µg of the TEAD4-ΔN-promoter–luciferase reporter plasmid. The amount of transfected DNA was kept constant at 2 μg by adding pUC18 plasmid. The relative fold-change was calculated based on the measurement taken from samples transfected with the TEAD4-ΔN-promoter–luciferase reporter plasmid only.

### 4.8. DNA Methylation Analysis—Bisulfite Sequencing (BS) [42]

The genomic DNA was prepared using the Quick-DNA Miniprep Kit (Zymo Research, Irvine, CA, USA) from umbilical cord and placenta tissue samples obtained from the Department of Obstetrics and Gynaecology at the University of Pécs. Genomic DNA preparation from human samples was approved by the Ethics Committee of the University of Pécs (Code: 3648—PTE 2020, Epigenetic and Transcriptional Factors Involved in Placental Development). The isolated DNA samples were then treated with bisulfite using the EZ DNA methylation kit (Zymo Research, Irvine, CA, USA). MethPrimer software v.2.0 was used to design primers for the BS. PCR-amplified promoter regions were cloned into the pDrive vector. Plasmids were purified from 5 bacterial colonies, and the methylation status of CpGs was determined by Sanger sequencing.

### 4.9. Western Blotting

The cells on a confluent plate measuring 100 mm were lysed using M-Per mammalian protein-extraction buffer (Thermo Scientific, Waltham, MA, USA) supplemented with a protease inhibitor (Roche, Basel, Switzerland) and phosphatase inhibitor (Sigma) cocktail. The umbilical cord and placenta tissue samples were obtained from the Department of Obstetrics and Gynaecology at the University of Pécs. Protein-extract preparation from human samples was approved by the Ethics Committee of the University of Pécs (Code: 3648—PTE 2020, Epigenetic and Transcriptional Factors Involved in Placental Development). Tissue samples were homogenized in M-Per mammalian protein-extraction buffer (Thermo Scientific) supplemented with a protease inhibitor (Roche) and phosphatase inhibitor (Sigma) cocktail using a Dounce homogenizer. The resulting lysates were loaded onto 12% SDS–polyacrylamide gels and transferred onto PVDF membranes (Amersham). Membranes were blocked in 5% milk and incubated overnight at 4 °C with primary antibodies against TEAD4 (1:200 final dilution; Thermo Scientific) and β-Actin (1:1000 final dilution; Cell Signaling). Horseradish peroxidase-conjugated secondary antibodies specific to the species were used at a final dilution of 1:2000 (Cell Signaling). The immunocomplexes were visualized using Immobilon ECL Ultra Western HRP Substrate (Merck) and a Syngene G:BOX Chemiluminescence and Fluorescence imaging system (Syngene). Results were analyzed with GeneSys software v.2.1 (Syngene). Bio-Rad Precision Plus Protein TM Standards Kaleidoscope TM was used as molecular-weight markers.

### 4.10. Electrophoretic-Shift Essay (EMSA)

To identify and characterize protein–DNA-binding interactions associated with the TEAD4-ΔN promoter, we used the LightShift EMSA kit (Thermo Scientific) and followed the suggested protocol. The biotinylated oligonucleotides are listed in Appendix A. The protein extract for the EMSA was prepared from 80% confluent HEK293 cell cultures as previously described [43].

### 4.11. Statistical Analysis

At least three independent experiments (triplicates) were conducted for all presented data. Data in the figures represent the mean ± SEM. Statistical differences were determined using the paired Student’s *t*-test or One-Way ANOVA with Tukey HSD and Mann–Whitney tests. The specific differences were considered statistically significant if *p* < 0.05. Statistical significance is indicated by asterisks as follows: *p* < 0.05 *; *p* < 0.01 **.

## Figures and Tables

**Figure 1 ijms-25-02223-f001:**
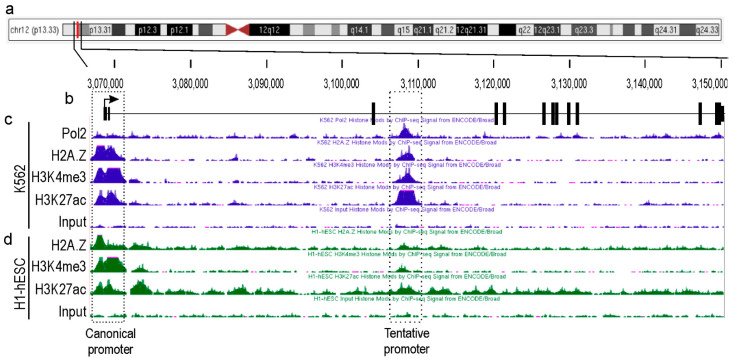
Epigenetic landscape of the TEAD4-gene-encoding locus in permanent cell lines. (**a**) Human chromosome 12. The vertical red line indicates the precise location of the TEAD4 gene on the chromosome. (**b**) Exon/intron structure of TEAD4 gene. The diagram indicates the exons by the thick vertical lines and the intronic regions by the thin horizontal lines. The direction of transcription is indicated by the arrow. The epigenetic histone and Pol2 ChIP-Seq profiles are shown for the K562 (**c**) and H1-hESC (**d**) cell lines.

**Figure 2 ijms-25-02223-f002:**
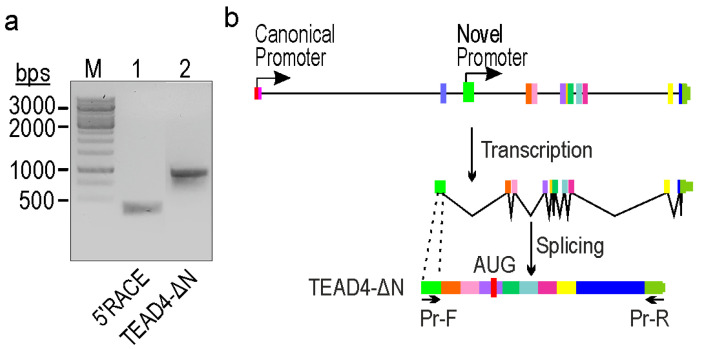
Identification of a novel TEAD4-isoform-encoding mRNA transcribed from an alternative intronic promoter. (**a**) 5′RACE analysis of the TEAD4 gene (lane 1) and the amplicon of the truncated form of the TEAD4 gene (lane 2). (M = Molecular weight marker). (**b**) TEAD4 gene’s exon/intron structure (exons are indicated by thick vertical lines of different colors; introns are indicated by thin horizontal lines), transcription, and splicing processes involved in the generation of the TEAD4-ΔN isoform. Pr-F and Pr-R indicate the position of the PCR primers used in the amplification of the novel transcript.

**Figure 3 ijms-25-02223-f003:**
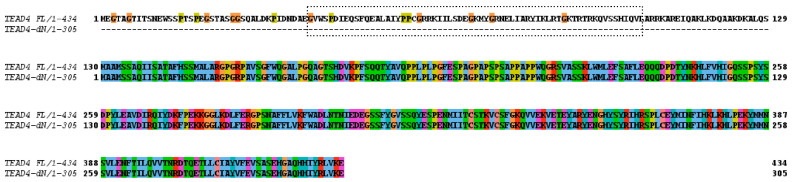
Pairwise protein-sequence alignment of TEAD4 isoforms. TEAD4-FL—amino acid sequence of the full-length TEAD4 protein. TEAD4-ΔN—amino acid sequence of the N-terminal DNA-binding-domain-deleted isoform. The DNA-binding domain is shown by the dotted frame, which is missing from the new TEAD4 isoform.

**Figure 4 ijms-25-02223-f004:**
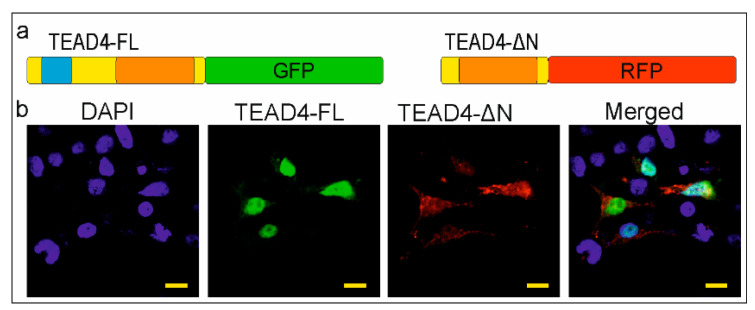
Subcellular localization of the TEAD4 isoforms. (**a**) Fusion constructs of the TEAD4 isoforms are depicted. The DNA-binding domains are highlighted as the light blue box, while the YAP-binding domains are represented by orange boxes. (**b**) The green fluorescent protein (GFP) and red fluorescent protein (RFP) are used as fusion tags. HEK293 cells were co-transfected with recombinant plasmids expressing either the full-length TEAD4 (TEAD4-FL) or truncated TEAD4 (TEAD4-ΔN) isoforms. Nuclear staining was performed using DAPI. The images were captured at a magnification of 800× with a scale bar of 10 μm.

**Figure 5 ijms-25-02223-f005:**
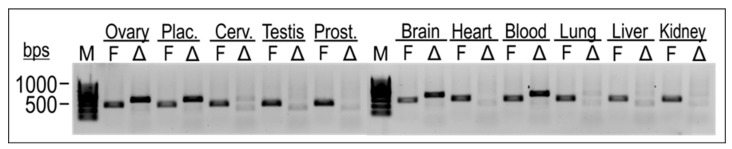
Isoform-specific end-point PCR was utilized to examine the expression of the TEAD4 isoforms in diverse tissue samples. RNA samples were reverse transcribed into cDNAs, and primer pairs specific for the full-length TEAD4 (TEAD4-FL) and truncated isoform (TEAD4-ΔN) were used in end-point PCR reactions. Amplicons were detected by agarose gel electrophoresis. F denotes TEAD4-FL, while Δ refers to TEAD4-ΔN. (M = Molecular weight marker).

**Figure 6 ijms-25-02223-f006:**
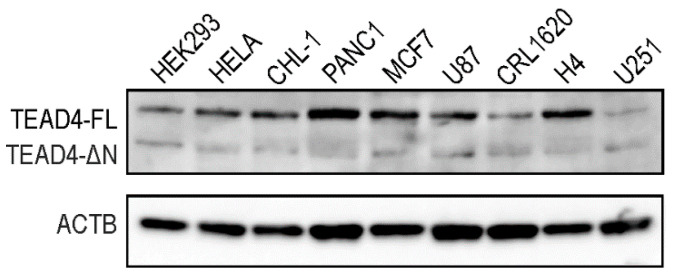
Expression levels of the TEAD4 proteins in cell lines. The expression pattern of full-length TEAD4 (TEAD4-FL) and truncated TEAD4 isoforms (TEAD4-ΔN) was detected by Western blotting. Human actin beta (ACTB) was used as a loading control.

**Figure 7 ijms-25-02223-f007:**
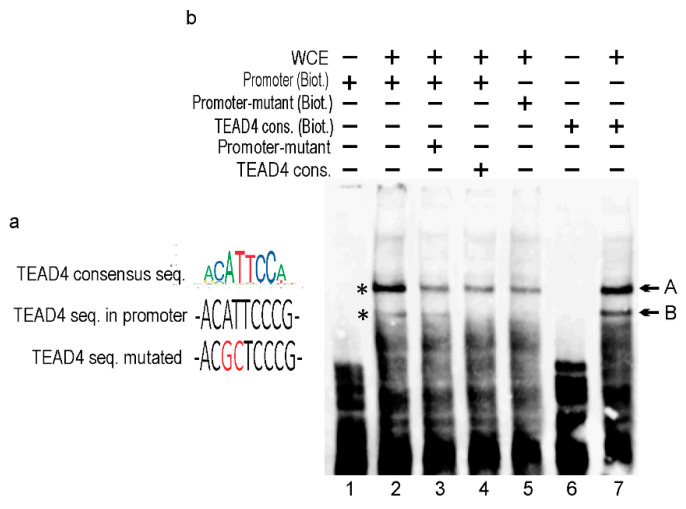
In vitro DNA–protein interaction study using EMSA. (**a**) The TEAD4 consensus motif is shown in color. The wild-type TEAD4 motif found in the TEAD4-ΔN promoter. The red label highlights the mutated core nucleotides in the TEAD4 binding site. (**b**) EMSA components included in each binding reaction are indicated using the +/− symbols. WCE—whole cell extract; Promoter (Biot.)—biotinylated promoter sequence containing the wild-type TEAD4 binding site; Promoter- mutant (Biot.)— biotinylated promoter sequence containing the mutant TEAD4 binding site; TEAD4 (Biot.)—biotinylated TEAD4 consensus sequence; Promoter-mutant and TEAD consensus—unbiotinylated double-stranded competitor oligonucleotides.

**Figure 8 ijms-25-02223-f008:**
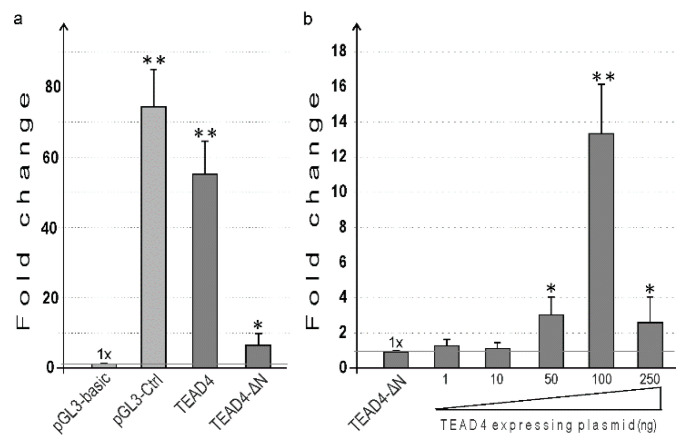
Functional characterization of the TEAD4 promoters in transient expression assays. Recombinant and control plasmids were transiently transfected, and luciferase activity was measured 48 h after transfection. (**a**) The TEAD4-ΔN promoter was functionally identified. The pGL3-basic plasmid was used as a promoterless control. The luciferase activity of the pGL3-basic plasmid-transfected samples was normalized to protein content and considered as 1-fold. pGL3-Ctrl: the plasmid used for transfection control is driven by the SV40 viral promoter and enhancer. There are two plasmids used for TEAD4: one carries a 1 kb long canonical promoter (TEAD4), while the other carries a 1.3 kb long intronic-region-inserted minigene construct (TEAD4-ΔN). (**b**) The effect of TEAD4 TF overexpression on the TEAD4-ΔN promoter-driven luciferase minigene. The luciferase activity of TEAD4-ΔN-plasmid-transfected samples was normalized to protein content (1-fold), and other luciferase activities were compared to that value. TEAD4-overexpressing plasmids were co-transfected at various concentrations with the TEAD4-ΔN reporter construct. Data sets show the mean ± SEM. * *p* ≤ 0.05; ** *p* ≤ 0.01.

**Figure 9 ijms-25-02223-f009:**
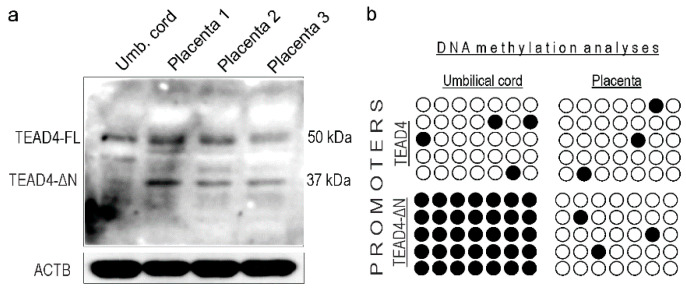
Protein expression and DNA methylation analyses were performed on human placental samples. (**a**) Western blot analysis of the expression of the TEAD4 isoforms in umbilical cord and three different placental samples. ACTB was used as a loading control. (**b**) Bisulfate sequencing analyses of TEAD4 promoters. Open circles represent unmethylated CpGs, while closed circles are methylated CpGs.

**Figure 10 ijms-25-02223-f010:**
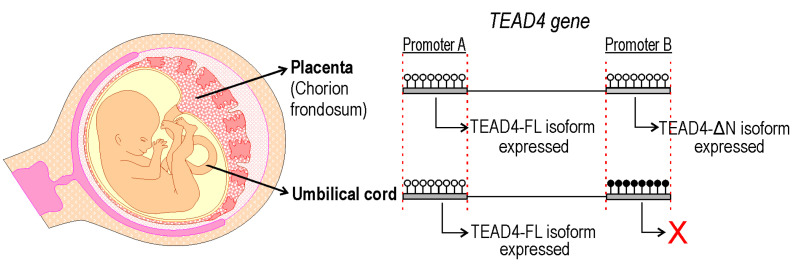
Epigenetic regulation of TEAD4 isoform expression in the placenta and umbilical cord. DNA methylation controls TEAD4 isoform expression. Promoter A, responsible for the expression of the full-length TEAD4 (TEAD4-FL), is unmethylated in all tissues, providing the constant presence of this isoform, while promoter B, responsible for the expression of the truncated isoform (TEAD4-ΔN), undergoes tissue-specific methylation, resulting in the tissue-specific presence of this isoform. Lollipops denote CpG dinucleotides: open ones are unmethylated; black-filled ones are methylated.

## Data Availability

Data are contained within the article.

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
