# Peer review of "Integrative Epigenetic and Molecular Analysis Reveals a Novel Promoter for a New Isoform of the Transcription Factor TEAD4"

_ijms, 2024, doi:10.3390/ijms25042223_

Round 1

Reviewer 1 Report

Comments and Suggestions for Authors

The methodology of this paper is rigorous, employing a variety of biological techniques such as 5’RACE, Western blotting, and dual-luciferase reporter assays, ensuring the reliability of the results. The combined application of these methods provides a solid experimental foundation for the discovery and functional study of TEAD4-ΔN. However, there are still some areas that could be optimized:

#1. The number of verification samples from the umbilical cord and placenta is too small to ascertain their representativeness.

#2. The paper mentions that TEAD4-ΔN could be beneficial for tumor treatment, but there is no discussion on its expression, methylation regulation, and functional role in cancer tissues.

#3. In terms of DNA methylation regulation, high methylation and low expression can be observed in umbilical cord tissues. It is recommended that, at the cellular level, methylation inhibitors be used to treat and observe changes in the expression of TEAD4-ΔN.

#4. In terms of format, the author divides the discussion and results into two separate sections. According to conventional practice, it is suggested that the results section should solely present findings without citing relevant literature. There is also the option to consider merging the results and discussion into one section as “Results and Discussion.”

Author Response

  1. Comments and Suggestions for Authors

The methodology of this paper is rigorous, employing a variety of biological techniques such as 5’RACE, Western blotting, and dual-luciferase reporter assays, ensuring the reliability of the results. The combined application of these methods provides a solid experimental foundation for the discovery and functional study of TEAD4-ΔN. However, there are still some areas that could be optimized:

Authors’ answer:

We appreciate the reviewer's time spent reviewing our manuscript and their constructive comments and suggestions. We have incorporated the suggested additions and discussions to make the manuscript clearer.

 Reviewer’s comment #1:

The number of verification samples from the umbilical cord and placenta is too small to ascertain their representativeness.

 Authors’ answer:

Before responding to this comment, we would like to provide some additional information that was not included in the original manuscript. Previously, it was unclear whether translational regulation was involved in TEAD4-ΔN expression in addition to the transcriptional regulation shown. We used RT-qPCR and Western blotting analysis to address this question.

FIGURE CAN BE FIND IN THE PDF COPY

Figure: TEAD4 expression in placenta and umbilical cord samples. Samples were monitored using isoform-specific RT-qPCR to follow the express pattern of the full-length (F) and deleted (D) TEAD4 isoforms. The expression level of the full-length TEAD4 isoform was set to 1, and measured CT values were normalized to ACTB expression. #5 –> 11: sample numbers; F = full length TEAD4, D = TEAD4-ΔN; n/d – not detectable transcript.

Western blot analysis detected the TEAD4-ΔN protein isoform in all samples where RT-qPCR detected mRNA encoding the alternative isoform. We concluded that (i) protein expression completely overlaps with mRNA expression, (ii) translational regulation is not involved in TEAD4-ΔN expression, and (iii) TEAD4-ΔN expression can be reliably monitored by either RT-qPCR or Western blotting. RT-qPCR is a less cumbersome method than Western blotting and provides more accurate quantitative data sets. Therefore, RT-qPCR was used in later experiments to monitor the expression of the TEAD4-ΔN isoform. To demonstrate the representativeness of the TEAD4 isoform expression pattern, we present here RT-qPCR data performed on 7 placental and corresponding umbilical cord samples. All samples investigated exhibited the same expression pattern. Specifically, the short isoform was exclusively expressed in placentas and was absent in umbilical cord samples. Therefore, the Western blotting data presented in Figure 9a is a representative experiment that illustrates the expression pattern of TEAD4-ΔN isoforms.

 Reviewer’s comment #2:

The paper mentions that TEAD4-ΔN could be beneficial for tumor treatment, but there is no discussion on its expression, methylation regulation, and functional role in cancer tissues.

Authors’ answer:

We have added some paragraph to the manuscript to address these particularly important issues. Accordingly, the Discussion part of the manuscript has been significantly reorganized. See colored text line 358 - .

 Reviewer’s comment #3:

In terms of DNA methylation regulation, high methylation and low expression can be observed in umbilical cord tissues. It is recommended that, at the cellular level, methylation inhibitors be used to treat and observe changes in the expression of TEAD4-ΔN.

 Authors’ answer:

The two most commonly used DNA methylation inhibitors are azacytidine and decitabine, both of which are nucleoside analogues that can be incorporated into genomic DNA using various enzymes of eukaryotic cells. We received placenta and umbilical cord samples after delivery and cells were not active to incorporate DNA inhibitors and conduct the suggested experiments. Therefore, the effect of DNA methylation inhibitors on the expression of TEAD4-ΔN in umbilical cord tissues could not be addressed in this experimental scenario.

 Reviewer’s comment #4:

In terms of format, the author divides the discussion and results into two separate sections. According to conventional practice, it is suggested that the results section should solely present findings without citing relevant literature. There is also the option to consider merging the results and discussion into one section as “Results and Discussion.”

Authors’ answer:

Presenting and discussing research data separately or together has advantages and disadvantages. In our original manuscript, we adhered to the research article template provided by the IJMS. If the editor suggests that the presentation could be made more readable for the audience, we may consider changing its form.

Reviewer 2 Report

Comments and Suggestions for Authors

In the present study, Rashidiani et al. have identified a novel promoter for the new isoform of TEAD4. The study is novel, well-designed, and may be useful for the researchers in the relevant field. I have a few minor concerns as follows:

1. What is the relevance of choosing H1-hESC and K562 cell lines in the study? Please discuss.

2. Is there any specific reason of choosing two different methods for analyzing TEAD4 expression in tissues and cells (as shown in Fig. 5 and 6)?

3. Please discuss how this discovery could have therapeutic application in the pathogenesis of diseases.

Author Response

  1. Comments and Suggestions for Authors

In the present study, Rashidiani et al. have identified a novel promoter for the new isoform of TEAD4. The study is novel, well-designed, and may be useful for the researchers in the relevant field. I have a few minor concerns as follows:

Authors’ answer:

We appreciate the reviewer's time spent reviewing our manuscript and their constructive comments and suggestions. We have incorporated the suggested additions and corrections to make the manuscript clearer.

Reviewer’s comment #1:

What is the relevance of choosing H1-hESC and K562 cell lines in the study? Please discuss.

Authors’ answer:

We added to the manuscript the reasons for using these two specific cell lines in our experiments. We concentrated on the analysis of the H1-hESC and K562 cell lines’ data sets because the ENCODE database revealed that the intronic region of the TEAD4 gene has such an epigenetic milieu that could support transcriptional competence.

See lines: 117 - 120

Reviewer’s comment #2:

Is there any specific reason of choosing two different methods for analyzing TEAD4 expression in tissues and cells (as shown in Fig. 5 and 6)?

Authors’ answer:

There are at least two reasons, one theoretical and one practical, for using two methods (i.e., end-point PCR and Western blotting) to study the expression pattern of TEAD4 isoforms.

Theoretical rationale: We wanted to show that the 5'RACE-identified mRNA version could be translated in the cells, since translational regulation may be involved in controlling the expression of a particular gene. Another possibility is that it is not translated and is "simply" a long non-coding RNA of unknown function. However, we could see that the 5'RACE identified mRNA version encodes a protein that is a truncated version of the full length TEAD4, since the anti-TEAD4 antibody recognized the encoded short isoform.

Practical rationale: We only had total RNA samples from healthy human tissues; therefore, Western blotting was excluded from the technical arsenal and only end-point or RT-qPCR could be used. Of note, we observed in preliminary studies that Western blot analysis detected the TEAD4-ΔN protein isoform only in samples where the mRNA encoding the alternative isoform was also detected by PCR. Therefore, a PCR-based approach is a simple and more rapid and quantitative method for monitoring the expression of the TEAD4-ΔN isoforms.

Reviewer’s comment #3:

Please discuss how this discovery could have therapeutic application in the pathogenesis of diseases.

Authors’ answer:

We have added a paragraph to the manuscript to address this particularly important issue. Accordingly, the “Discussion” part of the manuscript has been significantly reorganized to provide the requested information.

Reviewer 3 Report

Comments and Suggestions for Authors

In this manuscript, the authors identified a previously unknown transcript isoform of TEAD4 via ChIP-Seq assay. Next, the authors found that the newly identified TEAD 4 transcript encodes a truncated isoform that lacks the DNA-binding domain and is mainly localized in the cytoplasm. Next, the authors found that TEAD4 interacts with the alternative promoter and increases the expression of the truncated isoform. Finally, the authors found that DNA methylation plays a crucial function in the restricted expression of the TEAD4-ΔN isoform in specific tissues. In general, the authors present a lot of interesting results, and my assessment is positive. However, I think some concerns should be addressed prior to publication in IJMS. 

Specific comments: 

1)    In Figure 4, can the authors quantify the cell number ratio that the truncated form of TEAD4 was mainly present in the cytoplasm?

2)    In figure 5, can the authors detect the protein expression of the full-length TEAD4 and truncated isoform in different tissue samples?

3)    In figure 9a, can the authors quantify the western blot results of TEAD4-FL and truncated TEAD4?

4)    In figure 9b, can the authors quantify the methylation ratio in different samples? 

Author Response

  1. Comments and Suggestions for Authors

In this manuscript, the authors identified a previously unknown transcript isoform of TEAD4 via ChIP-Seq assay. Next, the authors found that the newly identified TEAD 4 transcript encodes a truncated isoform that lacks the DNA-binding domain and is mainly localized in the cytoplasm. Next, the authors found that TEAD4 interacts with the alternative promoter and increases the expression of the truncated isoform. Finally, the authors found that DNA methylation plays a crucial function in the restricted expression of the TEAD4-ΔN isoform in specific tissues. In general, the authors present a lot of interesting results, and my assessment is positive. However, I think some concerns should be addressed prior to publication in IJMS. 

We appreciate the reviewer's time spent reviewing our manuscript and their constructive comments and suggestions. We have incorporated the suggested additions and corrections to make the manuscript clearer.

 Reviewer’s comment #1:

1) In Figure 4, can the authors quantify the cell number ratio that the truncated form of TEAD4 was mainly present in the cytoplasm?

 Authors’ answer:

In transient transfection studies, we found that the presence of TEAD4-ΔN:RFP fusion protein could be detected in 96% of the cytoplasm. Notably, this number was highly dependent on the efficiency of co-transfection, but was never higher than 4% in nuclei.

 Reviewer’s comment #2:

2) In figure 5, can the authors detect the protein expression of the full-length TEAD4 and truncated isoform in different tissue samples?

 Authors’ answer:

Our studies revolve around embryonic implantation into the uterus and placental formation, so we have focused primarily on tissues related to human reproduction, along with some control tissues such as brain, liver and lung. However, some ongoing projects are broadening this scope to include more human tissues.

 Reviewer’s comment #3:

3) In figure 9a, can the authors quantify the western blot results of TEAD4-FL and truncated TEAD4?

We have found that PCR and Western blotting data have a good correlation, they verify each other. Therefore, we preferred to use PCR (i.e., RT-qPCR) for quantitative analysis because it is more user-friendly and provides more accurate quantification.

Reviewer’s comment #4:

4) In figure 9b, can the authors quantify the methylation ratio in different samples?

Using BS-sequencing approach, we found that the novel TEAD4-ΔN promoter was heavily (in 100%) methylated in umbilical cord samples implicating that epigenetic regulation via DNA methylation takes place. The same promoter (i.e., TEAD4-ΔN region) in placenta and the promoter of the full-length TEAD4 isoform was just sporadically methylated. There were no preferred CpG sites to be methylated implying this is just a random event in these promoters, and can be considered as just background DNA methylation.

Round 2

Reviewer 1 Report

Comments and Suggestions for Authors

Encourage the author to further experimentally validate the functionality and regulatory mechanisms.

Reviewer 3 Report

Comments and Suggestions for Authors

accept